# Evaluation of an autonomous acoustic surveying technique for grassland bird communities in Nebraska

Grace E. Schuster[1]*, Leroy J. Walston[2], Andrew R. Little[1]

1 School of Natural Resources, University of Nebraska-Lincoln, Lincoln, Nebraska, United States of America,
2 Environmental Science Division, Argonne National Laboratory, Lemont, Illinois, United States of America

* gschuster@huskers.unl.edu

## Abstract

Monitoring trends in wildlife communities is integral to making informed land management decisions and applying conservation strategies. Birds inhabit most niches in every environment and because of this they are widely accepted as an indicator species for environmental health. Traditionally, point counts are the common method to survey bird populations, however, passive acoustic monitoring approaches using autonomous recording units have been shown to be cost-effective alternatives to point count surveys. Advancements in automatic acoustic classification technologies, such as BirdNET, can aid in these efforts by quickly processing large volumes of acoustic recordings to identify bird species. While the utility of BirdNET has been demonstrated in several applications, there is little understanding of its effectiveness in surveying declining grassland birds. We conducted a study to evaluate the performance of BirdNET to survey grassland bird communities in Nebraska by comparing this automated approach to point count surveys. We deployed ten autonomous recording units from March through September 2022: five recorders in row-crop fields and five recorders in perennial grassland fields. During this study period, we visited each site three times to conduct point count surveys. We compared focal grassland bird species richness between point count surveys and the autonomous recording units at two different temporal scales and at six different confidence thresholds. Total species richness (focal and non-focal) for both methods was also compared at five different confidence thresholds using species accumulation curves. The results from this study demonstrate the usefulness of BirdNET at estimating long-term grassland bird species richness at default confidence scores, however, obtaining accurate abundance estimates for uncommon bird species may require validation with traditional methods.

## Introduction

Effective monitoring of wildlife populations is necessary to understand population trends and implement management strategies. A fundamental goal in most wildlife monitoring programs is to collect as much information on wildlife occurrence while controlling for biases in the

---

**Data Availability Statement:** All relevant data are within the manuscript and its Supporting Information files.

**Funding:** GES, LJW, and ARL received a Department of Energy grant to fund this work. The

specific grant number is DE-EE0009279. No commercial companies funded this study or any of the authors. This is the URL to the funder website: https://www.energy.gov/eere/bioenergy/bioenergy-technologies-office The funders did not play a role in study design, data collection and analysis, decision to publish, or preparation of the manuscript.

**Competing interests:** The authors have declared that no competing interests exist.

monitoring data associated with the influence of human survey efforts [1]. Advancements in wildlife monitoring in the past century, with the development of emerging technologies, offer new opportunities to expand ecological research [2]. These methods have allowed ecologists to record the effects of a rapidly changing environment on biodiversity. Among these technologies, the trending use of passive acoustic monitoring (PAM) for avian research has proven to be a non-invasive approach to long-term monitoring [3–5], which can be used to achieve several monitoring objectives.

Compared to traditional avian survey methods (e.g., point count surveys), PAM using autonomous recording units (ARUs) can be advantageous by minimizing human disturbances that could influence the monitoring data [6], potentially generating large amounts of data over space and time that can be analyzed at current or future time periods [7, 8], and answering unique management questions through increased frequency and duration of monitoring observations [9–11]. In addition, acoustic recordings are a perpetual dataset so they can be re-examined and reinterpreted as new questions arise or when interested in a different species [12]. Despite the many advantages to using PAM, there are obstacles that can limit the prospect of estimating population characteristics. PAMs focus on vocalizing animals only; therefore, an important assumption is made that landscape level biotic sound diversity reflects overall community diversity [13, 14]. Fundamental ecological parameters such as species abundance and overall population size can also be challenging to estimate from sound recordings. Although methods to infer densities from recordings have been increasingly applied in recent years [4]. During the breeding season, many grassland birds, specifically passerines, increase vocalizations to attract mates and defend their territory. However, these communication cues are not consistent across bird groups, thus another limitation is detecting cryptic species that may have lower detection probabilities because they do not vocalize frequently or loudly [6, 15]. Finally, massive amounts of audio data may be generated with long-term ARU deployment, requiring researchers to develop a specified approach to efficiently process the data. Early studies that used PAMs depended on trained professionals to manually identify bird songs and calls to annotate the recordings [16]. The annotation process can require many trained person-hours, so subsamples of the data are chosen to shorten the time it would take to manually annotate the recordings by determining if a species is present at the sample site. Recent developments have leveraged the use of artificial intelligence (AI) to expedite the process to identify bird species from acoustic recordings. One novel application, BirdNET, has been recently launched that can automatically classify bird species based on their recorded vocalizations [17]. Automatic classifiers such as BirdNET have the capability to process long recordings quickly and can increase the probability of detecting the presence of rare or obscure individuals that may be absent in the subsampled data.

Developed in a joint effort between the Cornell Lab of Ornithology and the Chemnitz University of Technology [10], BirdNET is an open access technology that can be operated as a Python script on a computer console, a mobile phone application, or used from an internet browser (https://birdnet/cornell.edu/). This automatic acoustic classifier uses convolutional neural network algorithms to identify bird vocalization in 3-second (s) segments of a longer audio recording [4]. With each prediction segment, BirdNET provides a confidence score ranging from 0 to 1 indicating how confident the program is that a species is present on the recording [18]. There are several parameters that could be used to generate species predictions through BirdNET such as overlapping of the 3-s prediction segments (overlapping segments can increase species detection outcomes), the sensitivity parameters (higher values may be more valuable in species-rich environments), and spatial and temporal filters [10, 19].

The current version of BirdNET (v2.4) has the capacity to identify >6,000 bird species worldwide [18]. As a result, BirdNET may be useful in evaluating bird occurrence patterns at

the target species or community levels. Despite this, few studies have used BirdNET to address wildlife management questions. Most studies using BirdNET have focused on evaluating Bird-NET performance in a variety of landscapes [5, 15, 20] and using a variety of recording devices [11]. Many of these studies have found that BirdNET performs with high precision rates for both focal (species-specific) or soundscape recordings and can be comparable to traditional monitoring methods (i.e., point count surveys). For example, Cole et al. [5] found that Bird-NET precision was >0.70 for most of the 13 bird species examined in that study; they also found that automatic species classifications using BirdNET from long-duration acoustic recordings produced occupancy model outputs that were similar to manually annotated data. Arif et al. [20] found that BirdNET had an average precision rate of 0.84 for a subset of detections with a confidence level of 0.95 or higher. Although raw audio data collected by ARUs typically contains vocalizations of many bird species [21], only a handful of studies have focused on evaluating BirdNET's ability to identify multiple species at the community-level (e.g. [15, 20]). This presents an opportunity to assess BirdNET's performance in correctly determining the presence of multiple bird species (species richness) compared to traditional survey methods such as point count surveys.

We conducted a study to evaluate the performance of BirdNET to survey grassland bird communities in Nebraska. The avian species included in this study are breeding grassland birds, but some species that were monitored are also year-round residents of Nebraska. This study is part of a larger project focusing on the influence of bioenergy crop production on grassland bird communities in agriculture landscapes. Grassland birds in North America have experienced a 53% population decline since the 1970s [22] due to loss of habitat associated with the conversion of grassland habitat to row-crop agriculture and declines in insect prey populations [23, 24]. Systematically obtaining accurate estimates of species richness is an important aspect of determining the effects of agriculture intensification on grassland bird communities. Our objectives were to (1) evaluate the performance of BirdNET in identifying focal grassland bird species at six confidence thresholds, including a species-specific confidence threshold, and (2) compare estimates of species richness determined from passive acoustic monitoring using BirdNET to those obtained from in-person point count surveys conducted during the same study period and at the same locations. Our comparisons were made at two monitoring durations: short-duration monitoring (using only acoustic recordings made within 1 hour (hr) of point count surveys) and long-duration monitoring (using all acoustic recordings made within the point count survey period). We also provide recommendations based on our results to inform future avian community-based PAM efforts using BirdNET.

## Methods

### Study area

Point count surveys were conducted, and ARUs were deployed on six grassland and six row-crop sites in Hayes and Hitchcock County, Nebraska (Fig 1). Each grassland site was paired with a neighboring row-crop site. The distance between each paired grassland-cropland site was between 500 and 1000 meters (m), and pairs were >8 kilometers (km) (5 miles) from one another. The landscape of southwestern Nebraska is dominated by agriculture fields, little bluestem-grama mixed prairie, and Sandhills upland prairie [25]. Most of the six grassland sites selected for this study were enrolled in the Conservation Reserve Program (CRP), which is implemented by the U.S. Department of Agriculture (USDA) Natural Resources Conservation Service (NRCS) [26] and were dominated by Big Bluestem (*Andropogon gerardii*), Little Bluestem (*Schizachyrium scoparium*), and Brome (*Bromus inermis)*. The row-crop sites

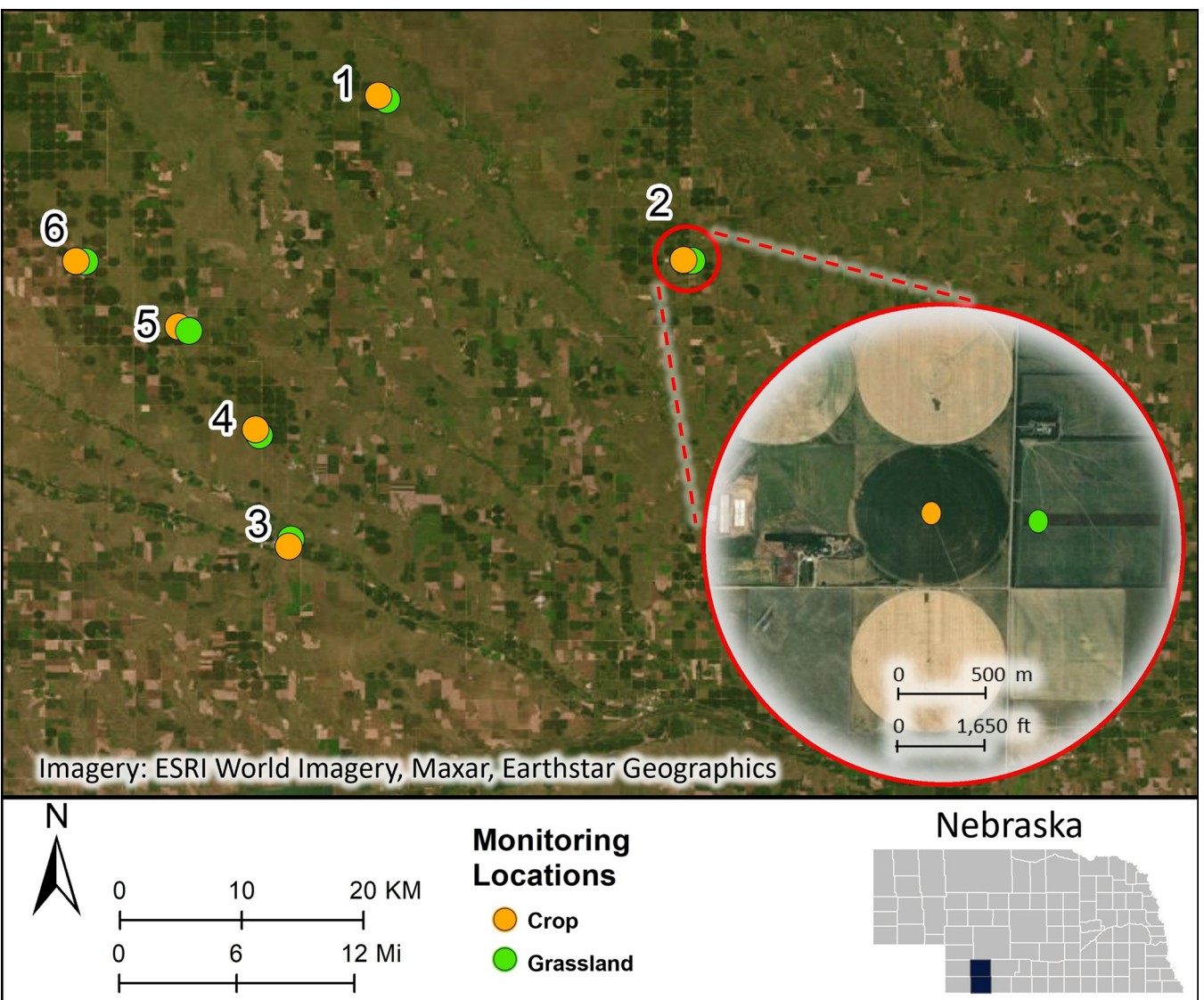

**Fig 1. Study area in southwestern Nebraska showing the locations of the six paired monitoring sites.** Map was created using ArcGIS (v10.8). Basemap satellite images obtained from the World Imagery Esri layer under a CC BY license. Sources: Esri, Maxar, Earthstar Geographics, and the GIS User Community. Content is the intellectual property of Esri and is used herein with permission. Copyright © 2024 Esri and its licensors. All rights reserved.

consisted of center-pivot irrigation row-crop fields, most of which were cultivated for corn (*Zea mays*) during this study. Sampling for this study took place from March–September 2022.

## Sampling design

**Point count surveys.** Point Count surveys took place between 19 May and 19 July 2022. Each site was visited three times during this period between 0600 and 1000 CST on days with no high winds (>16km) or precipitation. We selected these dates for sampling because it encompasses most grassland birds' breeding season (mid-May to early August) [27], and the time of day is when bird vocalizations are most frequent [28]. At each visit, observers trained in the identification of Nebraska grassland birds conducted a 10-minute survey [29] at random points within the grassland fields and at the pivot center in the row-crop fields. Pivot centers

are the center point of a circular row-crop field, and they consist of an open area that is not planted with crops. Point counts were not completed on row-crop sites when pivots were operating to limit excessive noise interference and protect the safety of observers. We collected information on individual birds detected, including the species, sex of the bird, estimated distance from the observer with no maximum detection radius, and behavior (singing, calling, or observed).

**Acoustic monitor deployment.**   Beginning on 17 March 2022, we deployed twelve ARUs (Wildlife Acoustics Song Meter Mini, Maynard, MA, USA), one at each grassland and row-crop site. The distance between ARUs at each paired field site was between 600 and 755 m. Evidence from past research has suggested that >100 m is an adequate detection distance for ARUs [30, 31]; however, detection space varies based on species of interest, study location, and ARU product [32]. The ARUs were programmed to record 16-bit WAV audio at a sample rate of 24000 Hz for a total of four hours each day: one hour before sunrise, one hour after sunrise, and one hour before sunset to one hour after sunset. The gain setting on the ARUs was set at the default setting (18 dB) and microphones were checked before installation. ARUs were installed at fixed locations in the grassland and row-crop fields. In each grassland, one ARU was installed in a random location >100 m (328 feet) from the edge of the field to ensure that the birds that were detected were within the study field. In the row-crop fields, the ARUs were installed at the pivot center to keep them from being damaged by farming equipment that might be used throughout the agriculture season. All recorders were affixed to a polyvinyl chloride (PVC) pole, which was secured to a metal fence post with stainless steel hose clamps and positioned approximately 2.5 m (8 feet) above the ground. The records all faced either North or South, while we did not face them all the same direction, recorder directionality should not have led to any bias in recording quality, or the number of species detected [33].

ARUs were visited approximately every two weeks to check batteries and digital memory (SD) cards until 12 September 2022, when they were removed from the field. We report results from 10 of the 12 sites because two ARUs had technological difficulties at one of the paired grassland and row-crop sites due to high winds. These high winds caused the SD cards to fall loose from the ARUs, presumably due to excessive movement and vibration, which led to several weeks of missing data from these ARUs.

## Evaluating BirdNET performance

**Recording analyses.**   Acoustic recordings collected by the ARUs were analyzed using the BirdNET-Analyzer graphical user interface (v2.2; available at https://github.com/kahst/BirdNET-Analyzer). We used the default parameter settings to run BirdNET: minimum confidence threshold of 0.10, sensitivity parameter of 1.0, and no (0) overlap of prediction segments. BirdNET was configured to make species predictions based on weekly eBird checklists for the geographic area near the center of all 12 monitoring locations (40.51˚ N, -101.02˚ W). We saved all BirdNET results as tab-delimited data tables for later analysis.

**Focal species selection.**   Based on the observations made during the point count surveys and review of the BirdNET detections, we selected 20 focal grassland bird species to prioritize for this study (Table 1). These focal species represented bird species commonly inhabiting Nebraska grasslands and were regularly detected during point count surveys or through PAM using BirdNET.

**Precision.**   We randomly selected 60 BirdNET-generated detections for each of the 20 focal species and generated 5-s audio segments for each detection. The 5-s audio segments included one additional second before and after the 3-s spectrogram detected by BirdNET to facilitate manual validation and annotation. An expert observer (GES or LJW) listened to each

**Table 1. Summary of BirdNET performance for 20 focal grassland bird species in Nebraska.** The maximum confidence score is the maximum confidence of all Bird-NET detections for each species. Precision ($P_{C10}$ and $P_{C25}$) was calculated based on validation of 60 random samples of BirdNET detections of each species. Recall was calculated from 50 randomly generated 120-s acoustic recordings. The F-Score represents the balanced measure of Precision (using $P_{C10}$) and Recall. The species-specific confidence threshold was determined from these validation results based on the logic presented in Table 2.

| Scientific Name | Common Name | Total Detections | Maximum Confidence Score | Default Precision ($P_{C10}$) | Precision ($P_{C25}$) | Max Confidence False Positive | Recall | F-Score | Species-Specific Confidence Threshold (SSC) |
|---|---|---|---|---|---|---|---|---|---|
| *Spinus tristis* | American Goldfinch | 4,004 | 0.999 | 0.967 | 1.000 | 0.186 | 1.000 | 0.983 | 0.100 |
| *Turdus migratorius* | American Robin | 75,153 | 0.999 | 0.933 | 0.939 | 0.551 | 0.500 | 0.651 | 0.100 |
| *Hirundo rustica* | Barn Swallow | 9,046 | 0.999 | 0.950 | 1.000 | 0.195 | — | — | 0.100 |
| *Passerina caerulea* | Blue Grosbeak | 26,670 | 0.999 | 0.683 | 0.879 | 0.914 | 1.000 | 0.812 | 0.250 |
| *Cyanocitta cristata* | Blue Jay | 6,539 | 0.999 | 0.983 | 1.000 | 0.206 | 0.667 | 0.795 | 0.100 |
| *Quiscalus quiscula* | Common Grackle | 8,407 | 0.998 | 0.867 | 0.939 | 0.574 | 1.000 | 0.929 | 0.100 |
| *Chordeiles minor* | Common Nighthawk | 26,179 | 0.998 | 1.000 | 1.000 | 0.100 | 1.000 | 1.000 | 0.100 |
| *Spiza americana* | Dickcissel | 397,405 | 0.997 | 0.933 | 1.000 | 0.200 | 0.588 | 0.722 | 0.100 |
| *Tyrannus tyrannus* | Eastern Kingbird | 28,748 | 0.997 | 1.000 | 1.000 | 0.100 | 1.000 | 1.000 | 0.100 |
| *Ammodramus savannarum* | Grasshopper Sparrow | 956,111 | 0.996 | 0.533 | 0.652 | 0.448 | 0.800 | 0.640 | 0.448 |
| *Eremophila alpestris* | Horned Lark | 581,287 | 0.996 | 0.817 | 0.955 | 0.628 | 0.833 | 0.825 | 0.250 |
| *Zenaida macroura* | Mourning Dove | 31,117 | 0.995 | 0.867 | 0.974 | 0.293 | 0.500 | 0.634 | 0.100 |
| *Colinus virginianus* | Northern Bobwhite | 34,608 | 0.995 | 0.817 | 1.000 | 0.227 | 1.000 | 0.899 | 0.250 |
| *Agelaius phoeniceus* | Red-winged Blackbird | 148,573 | 0.997 | 0.967 | 1.000 | 0.129 | 0.600 | 0.740 | 0.100 |
| *Phasianus colchicus* | Ring-necked Pheasant | 13,152 | 0.999 | 0.967 | 0.962 | 0.717 | 0.760 | 0.851 | 0.100 |
| *Passerculus sandwichensis* | Savannah Sparrow | 4,439 | 0.998 | 0.567 | 0.889 | 0.494 | 0.500 | 0.531 | 0.250 |
| *Tachycineta bicolor* | Tree Swallow | 7,637 | 0.996 | 0.117 | 0.333 | 0.505 | — | — | 0.505 |
| *Tyrannus verticalis* | Western Kingbird | 4,958 | 0.999 | 0.917 | 0.959 | 0.711 | — | — | 0.100 |
| *Sturnella neglecta* | Western Meadowlark | 1,331,498 | 0.999 | 0.650 | 0.821 | 0.598 | 0.867 | 0.743 | 0.250 |
| *Meleagris gallopavo* | Wild Turkey | 4,083 | 0.999 | 0.800 | 0.923 | 0.653 | 1.000 | 0.889 | 0.250 |

5-s segment and inspected the spectrogram to confirm whether the focal species was present or absent. This information was used to calculate precision, as follows:

$$\frac{true\ positives}{true\ positives + false\ positives}$$

For precision calculations, a *true positive* detection occurred when the observer confirmed that a species was accurately classified by BirdNET. A *false positive* detection occurred when BirdNET detected a species that was not confirmed by the observer. Precision was calculated twice: once for a random sample of 60 unfiltered BirdNET detections using default settings (e.g., minimum confidence threshold of 0.10; $P_{C10}$) and once for a random sample of 60

BirdNET detections filtered to a minimum confidence threshold of 0.25 ($P_{C25}$). From these precision calculations, we also estimated the maximum confidence score for false-positive detections for each focal species.

**Recall.**   We generated a random sample of 50 120-s acoustic segments from the total amount of ARU recordings. We used these 50 segments to compare BirdNET-derived focal species detections to manually annotated focal species detections of the same file. After running these segments through BirdNET using default parameters and a confidence threshold of 0.10, an expert reviewer (GES) listened to all 120-s acoustic files to manually annotate all focal bird species heard (n = 50 files). This information was used to calculate recall, as follows:

$$\frac{true\ positives}{true\ positives + false\ positives}$$

where true positives were the number of 120-s recordings that had at least one manual detection and at least one BirdNET detection of a given focal species. False negatives were 120-s recordings with manual detection but no BirdNET detection.

**F-Score.**   We calculated the F-Score as the harmonic mean of Precision and Recall as follows:

$$\frac{2[precision*recall]}{[precision + recall]}$$

The F-Score was interpreted as a balanced measure of precision and recall. We calculated the F-score using $P_{C10}$ precision calculations.

## Comparing BirdNET to point count surveys

To compare BirdNET's detection of the 20 focal species to traditional point count surveys, we first filtered all BirdNET detections based on six different confidence thresholds: 0.1, 0.25, 0.5, 0.75, 0.9, and a species-specific confidence (SSC) threshold. We determined the SSC threshold based on the BirdNET validation evaluation for each species (Table 2). If BirdNET's precision at the default confidence setting of 0.10 ($P_{C10}$) was greater than or equal to 0.85, then the default confidence threshold (0.10) was applied, and all detections were assumed valid. For any focal species with $P_{C10}$ estimates less than 0.85 but precision at a minimum confidence threshold of 0.25 ($P_{C25}$) greater than or equal to 0.85, the SSC threshold was set to 0.25. For all other focal species, the maximum confidence score for false-positive detections was used as the SSC threshold.

Species richness for the 20 focal species was calculated at each of the study sites for the point count surveys and ARUs. Focal species richness from the point count surveys was calculated as the total number of unique focal species seen or heard across the three survey visits at each location. Focal species richness from the ARUs was calculated as the unique number of focal species detected by BirdNET after filtering the BirdNET output to the six confidence thresholds described above. In addition, species richness comparisons between point count

**Table 2.  The BirdNET validation evaluation we used to determine species-specific confidence (SSC) thresholds for the 20 focal grassland bird species.**

| Precision estimate | Guideline |
|---|---|
| $P_{C10} \geq 0.85$ | No filter was applied (i.e., all BirdNET detections were assumed valid). |
| $P_{C10} < 0.85$ & $P_{C25} \geq 0.85$ | A confidence threshold of 0.25 was applied. |
| $P_{C10} < 0.85$ & $P_{C25} < 0.85$ | The Maximum Confidence of False Positives was applied. |

surveys and ARUs were conducted at two different monitoring periods: a short-duration and a long-duration monitoring period. Short-duration measures of focal species richness were calculated at each site based on the unique number of focal species detected by BirdNET from acoustic recordings within 1-hr of each point-count survey. While the ARUs were not always recording at the exact time when point count surveys were being conducted, most of them were within 1-hr apart, making the BirdNET short-duration focal species richness calculations the most appropriate instantaneous point of comparison with point count surveys. Second, a long-duration measure of focal species richness was calculated at each site based on the total number of focal species detected by BirdNET from all acoustic recordings during the 60 days that point count surveys took place (19 May 2022–19 July 2022). The long-duration focal species richness calculations were compared to the season-total focal species richness calculations made through the point count surveys.

In addition to comparing focal species richness, we compared total species richness estimates over the 60-day point count survey period. To do this, we calculated the total season-long species richness from point count surveys as the total number of unique species (focal species and non-focal species) observed or heard during point count surveys across all sites. We also identified the total number of bird species detected by BirdNET at each site and manually reviewed the associated spectrograms and 5-s acoustic segments around each BirdNET detection to determine whether any of the BirdNET detections could be confirmed. To expedite this process across the large volume of BirdNET detections, we ranked all species detections by BirdNET confidence score and reviewed the ten detections with the highest confidence scores for each species detected. Only species that were confirmed at least once among the ten highest confidence recordings were included in the ARU-based calculation of species richness. Species that could not be confirmed after listening to the ten highest confidence recordings were excluded.

We compared short-duration and long-duration focal species richness between point count surveys and the ARUs using 12 Student paired, two-tailed t-tests for the six different BirdNET confidence thresholds. We visually compared the focal species accumulation at the short-duration and long-duration measurements using species accumulation curves (SAC). Accumulation curves are long-standing tools to measure species richness [34]. We also prepared SACs for total species richness estimates, including both focal and non-focal species. The ARU-based SAC for all total species was created using five confidence thresholds (0.1, 0.25, 0.5, 0.75, and 0.9). Because BirdNET validation was not performed for all species, the SSC was not used in the SAC for total species richness. The paired t-test analysis was completed in R-studio v4.0.5 using the function "t.test" in base R [35]. All SACs were generated in R using the 'ggplot2' package v3.5.0 [36]. Processing of BirdNET output tables was conducted in R using the 'dplyr' package v1.1.4 [37]. Extraction and processing of acoustic files was conducted in R using the 'av' package v0.9.0 [38].

## Results

A combined total of 5,189 hours of acoustic files were recorded by the ten ARUs used in this study across the 5-month monitoring period (17 March 2022–12 September 2022). From these recordings, BirdNET made over 4,604,000 species detections under default settings. The 20 focal species made up 3,699,614 (80%) of the total detections. The three focal species most frequently detected by BirdNET were Western Meadowlark (*Sturnella neglecta)*, Grasshopper Sparrow (*Ammodramus savannarum*), and Horned Lark (*Eremophila alpestris*) (Table 1). Point count surveys detected a total of 41 species during the three separate visits to each location over a 60-day period between 19 May and 19 July (S1 Table in S1 File). The species

detected at the greatest percentage of sites during point count surveys were Western Meadowlark (100% of sites), Red-winged Blackbird (*Agelaius phoeniceus*) (100% of sites), Mourning Dove (*Zenaida macroura*) (100% of sites), Grasshopper Sparrow (80% of sites), Eastern Kingbird (*Tyrannus tyrannus*) (80% of sites), and Dickcissel (*Spiza americana*) (80% of sites). Savannah Sparrow (*Passerculus sandwichensis*) and Common Nighthawk (*Chordeiles minor)* were the only two focal species that were never detected with point count surveys at any of the sites.

### Performance metrics for focal grassland bird species

Using the 60 5-s acoustic clips corresponding to the randomly selected BirdNET detections, we calculated precision, maximum confidence of false positives, recall, and the F-score for the 20 focal bird species (Table 2). $P_{C10}$ was >0.85 for 12 of the 20 focal species (mean = 0.817) and ranged between a low of 0.117 for Tree Swallow and 1.000 for Common Nighthawk and Eastern Kingbird. $P_{C25}$ was >0.85 for 17 of the 20 focal species (mean = 0.911) and ranged from a low of 0.333 for Tree Swallow and 1.000 for several species including Red-winged Blackbird, Northern Bobwhite (*Colinus virginianus)*, Eastern Kingbird, Dickcissel, Common Nighthawk, Blue Jay (*Cyanocitta cristata*), Barn Swallow (*Hirundo rustica*), and American Goldfinch (*Spinus tristis*). The maximum confidence score for false-positive detections was below 0.50 for 11 of the 20 focal species (mean = 0.421) and ranged from 0.100 for Common Nighthawk and Eastern Kingbird to 0.914 for Blue Grosbeak (*Passerina caerulea*).

We manually annotated 50 random 120-s acoustic clips to calculate recall and understand if any focal species undetected by BirdNET were manually detected in any of the recordings. We manually detected 31 total grassland bird species including 17 of the focal species from these acoustic clips. At the default settings, BirdNET detected 51 total species including 18 focal species from these same acoustic files (S2 Table in S1 File). Recall for all but three focal species averaged 0.801 (range: 0.500–1.000) meaning that, on average, BirdNET was 80% consistent with manual annotation in detecting the focal species from these 120-s acoustic clips. Recall could not be calculated for three focal species; Western Kingbird (*Tyrannus verticalis)*, Barn Swallow, and Tree Swallow (*Tachycineta bicolor)* because these species were either not manually detected or not detected by BirdNET in the 120-s acoustic clips. Several species had perfect recall scores, including American Goldfinch, Common Grackle (*Quiscalus quiscula*), Blue Grosbeak, Common Nighthawk, Eastern Kingbird, Northern Bobwhite, and Wild Turkey (*Meleagris gallopavo*) (Table 1).

Across all focal species, the F-score had a mean of 0.802 and ranged from 0.634 for Mourning Dove to 1.00 for several species including Common Nighthawk, Eastern Kingbird, and Northern Bobwhite (Table 1). We did not calculate the F-score for Tree Swallow, Western Kingbird, or Barn Swallow because we could not calculate recall for these species.

### Comparing BirdNET to point count surveys

In the short duration (i.e., using only acoustic recordings within 1-hr of point count surveys), we generated 24 pairs of point count surveys and 60-min acoustic recordings from the ARUs. Six-point count surveys did not have corresponding ARU recordings within 1-hr of the survey; these six surveys were excluded from the short duration comparisons. There was no significant difference between the two methods when the BirdNET output was filtered by the SSC, 0.1, or 0.25, (Table 3). However, the point count surveys yielded significantly higher short duration estimates of focal species richness compared to more conservative BirdNET confidence thresholds of 0.5, 0.75, and 0.9 (Table 3). For example, in the short duration, point count surveys

**Table 3. Comparison of grassland bird focal species richness estimated by acoustic recording units (ARUs) and point count surveys.** Species richness was calculated across three time periods with ARUs using six different BirdNET confidence threshold settings. These ARU-based measures of species richness were compared to species richness calculated from point count surveys for the same time period. Positive average Δ species richness values indicate greater species richness calculated from the ARUs with BirdNET compared to the point count surveys. A * denotes statistical significance (α = 0.05) using paired t-tests. See Methods for description.

| Confidence Threshold | Short Duration | Long Duration |
|---|---|---|
| Species-Specific Confidence (SSC) threshold | 0.208 | 8.9* |
| 0.1 | 1.375 | 9.1* |
| 0.25 | -0.375 | 8.5* |
| 0.5 | -1.5* | 8* |
| 0.75 | -2.25* | 7.2* |
| 0.9 | -3* | 6* |

identified an average of three more focal species per location than BirdNET using the most conservative confidence threshold of 0.9 (Table 3).

In the long duration (i.e., using all acoustic recordings and point count surveys in the 60-day period), we generated 1,879 hours of acoustic recordings across all five pairs of sites. BirdNET generated significantly greater ARU-based long-duration estimates of species richness at all confidence thresholds compared to the point count surveys (Table 3). When using the SSC threshold, BirdNET detected nearly nine more focal species per site. At the most conservative confidence threshold (0.9), BirdNET detected six more focal species per site at the long duration period (Table 3).

Cumulative focal species richness across all sites for each method are shown in Fig 2. At the short duration, neither ARUs nor point counts cumulatively detected all 20 focal species after the 60-day point count survey period (Fig 2A). BirdNET detected a maximum of 17 focal species from the ARUs using a BirdNET confidence threshold of 0.1. Although this cumulative total was reached within 20 days, the cumulative total focal species richness from the point count surveys eventually surpassed all ARU-based cumulative species richness estimates after 55 days (Fig 2A).

Not all 20 focal species were detected across all locations at the long-duration scale, using the ARUs or point counts. BirdNET detected 19 focal species within 20 days regardless of confidence thresholds, while the point counts yielded a slower rate of cumulative species accumulation that did not increase beyond 16 species until after 55 days (Fig 2B). Savannah Sparrow was the only focal species that was not detected by BirdNET or point count surveys during the 60-day point count survey period. Although we did not detect Savannah Sparrow during the 60-day point count survey period, we made over 4,400 BirdNET detections of this species before and after this time period.

We confirmed 95 total species (focal and non-focal) from the ARUs over the 5-month study period based on review of the ten BirdNET detections with the highest confidence scores for each species (S3 Table in S1 File). Of these 95 confirmed species, BirdNET detected between 55 and 74 at confidence levels of 0.9 and 0.1, respectively, during the 60-day point count survey period (Fig 3). Regardless of confidence threshold, the ARU-based cumulative total number of species detections exceeded the cumulative number of 41 total species (focal and non-focal) identified in the point count surveys by the end of the 60-day point count survey period.

## Discussion

Birds are an important indicator of ecosystem integrity because they are sensitive to changes in environmental conditions [39, 40], and they can be found in almost every niche in every

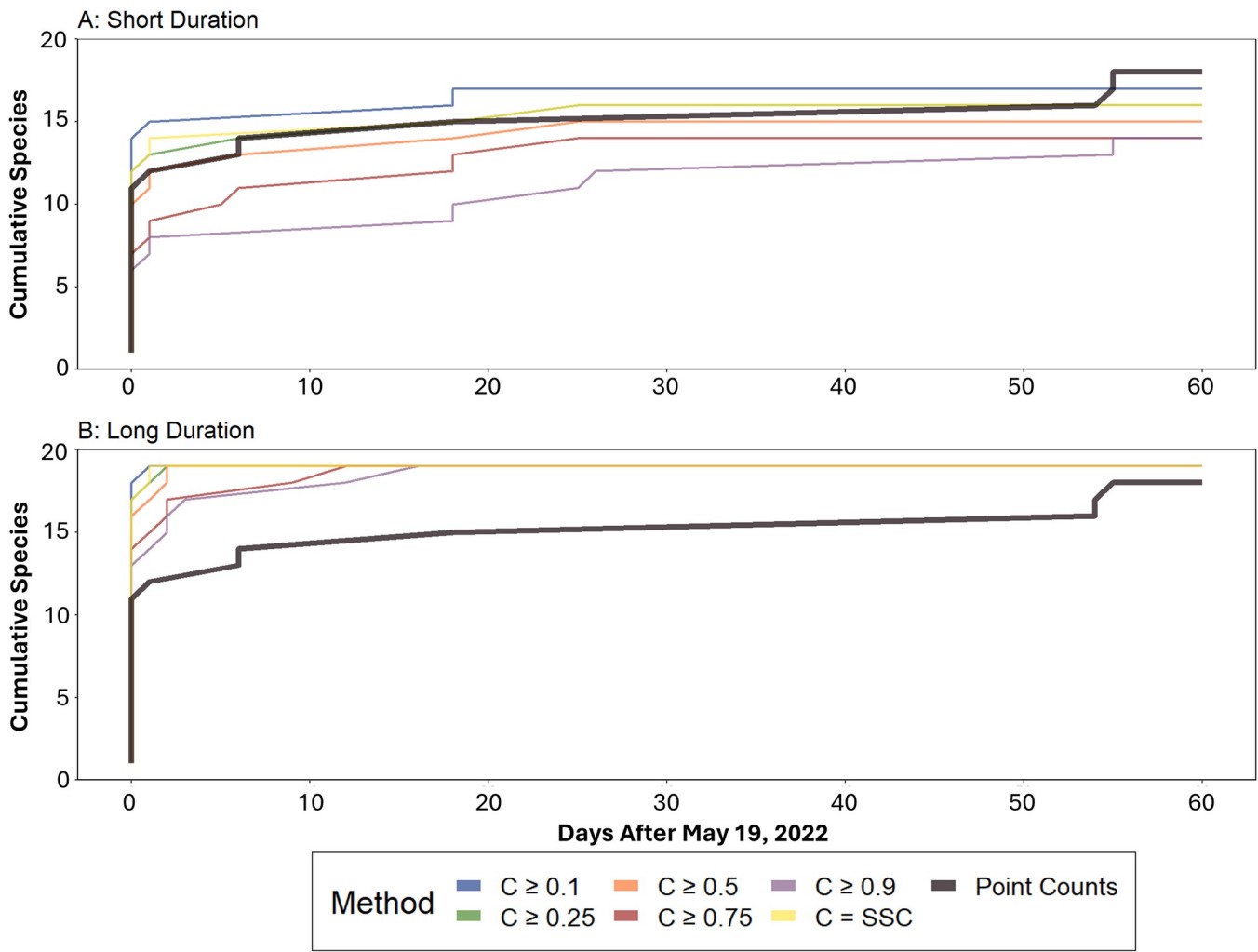

**Fig 2. Comparison of short-duration and long-duration species accumulation for point count surveys and ARUs at six different confidence thresholds (0.1, 0.25, 0.5, 0.75, 0.9, and SSC) from May 19th, 2022- July 19th, 2022.** Panel A depicts the short-duration comparison including point count surveys that had a paired acoustic recording within 1-hr of the survey. Panel B depicts the long-duration comparison including all point count surveys and recordings within the 60-day period that point counts took place.

environment [10]. Seven hundred million individual grassland birds across 31 species have been lost since the 1970s [22] due to conversion of grassland habitat to agriculture production. In the North American Great Plains region, recognized as the most extensive grassland system in North America spanning nearly 377 million acres across Canada and the United States, approximately 32 million acres of grasslands have been converted to agriculture since 2012 [41]. In Nebraska, 125,000 ha (309,000 acres) of grassland was converted to row-crop agriculture from 2006–2011 [42]. Given the rapid rate of land use conversion across large expanses of grassland systems in the U.S. and globally, it is critical to develop avian monitoring approaches that can be scalable and yield timely results to inform grassland bird management decisions. PAM with ARUs has emerged as a novel technology to meet these management needs. Despite the advantages of PAM over traditional methods (e.g., point count surveys), this new method generates large quantities of data, and procedures for analyzing these voluminous datasets are partially unknown [19, but see 15]. To address this information gap, we developed a framework for evaluating the performance of BirdNET in measuring avian species richness for an

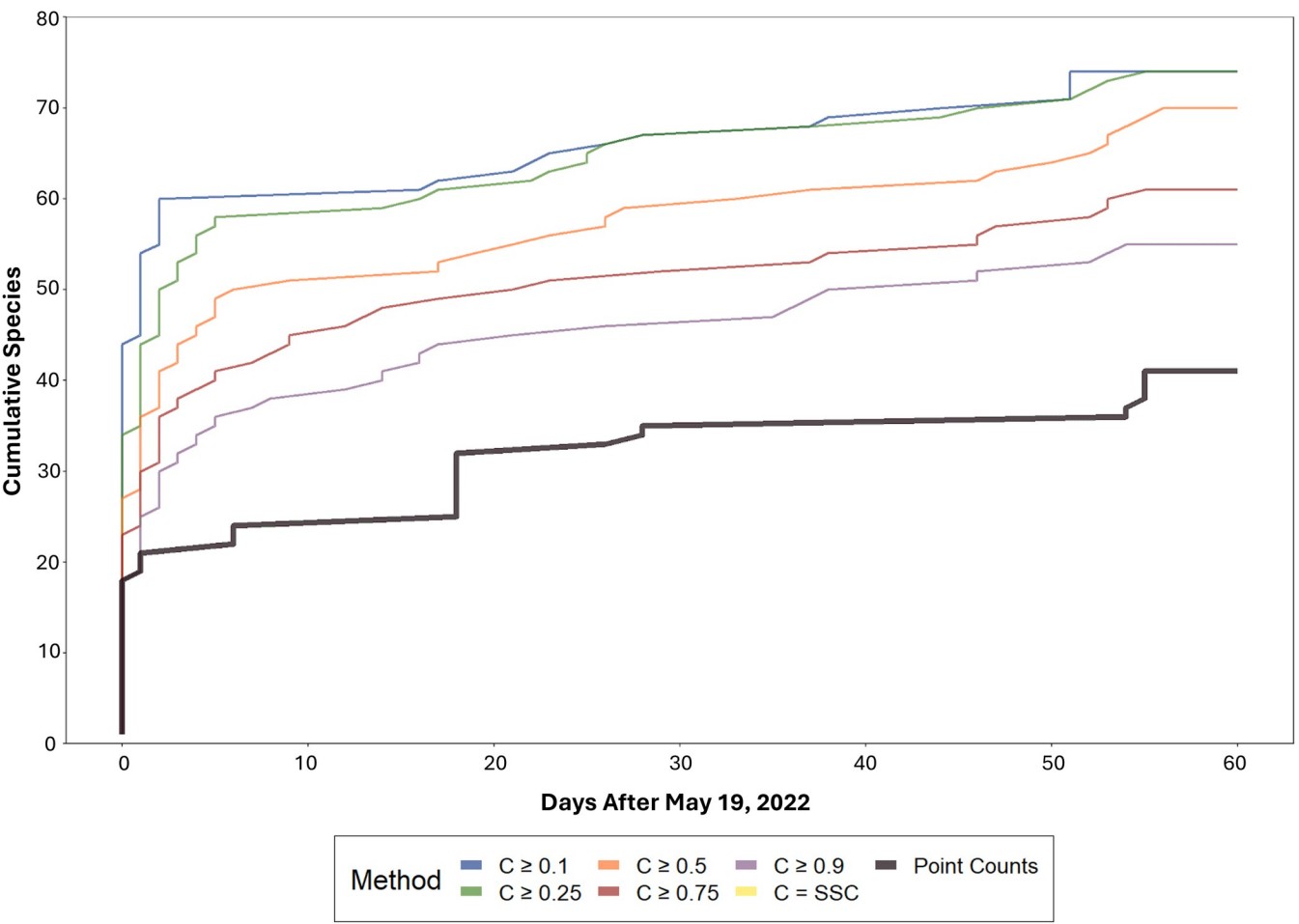

**Fig 3. Comparison of total species (focal and non-focal) accumulation from May 19th, 2022- July 19th, 2022, at five confidence thresholds (0.1, 0.25, 0.5, 0.75, and 0.9).** We did not include a species-specific confidence (SSC) threshold accumulation curve because the SSC was not calculated for all species.

assemblage of 20 focal grassland bird species and total species richness (focal and non-focal species) in Nebraska and compared these measures to those obtained from traditional point count surveys.

## Performance metrics for focal grassland bird species

Our results suggest that BirdNET performed effectively for most of our 20 focal species. Precision was ≥80% for 15 out of the 20 species at the default confidence threshold (0.10) and the overall average F-score was >80%. Blue Grosbeak, Grasshopper Sparrow, Tree Swallow, Western Meadowlark, and Savannah Sparrow were the only focal species that had default precision estimates ($P_{C10}$) less than 80% (Table 1). When listening to the acoustic clips for these five species during the validation process, we found that many of the false positives were misidentified by BirdNET due to confusion with other species with similar vocalizations. Extremely versatile species with large repertoires have a higher chance of confusion with other species that could affect performance scores [10]. Our results concur that BirdNET is an efficient multi-species recognizer [10, 17], however, we found several misidentification errors that were confirmed by manual validation. For example, several BirdNET detections for Blue Grosbeak were American

Robin (*Turdus migratorius)* vocalizations. In addition, many Grasshopper Sparrow detections were insect stridulations in the background of the recording.

Although the average recall for 17 out of 20 focal species was 80%, we were unable to calculate the performance metric for three species (Western Kingbird, Barn Swallow, and Tree Swallow) because they were not detected by manual annotation or detected by BirdNET in the 120-s acoustic clips. Low recall estimations could be problematic if the purpose of a study is to estimate animal density or evaluate behavior [19], but a low recall metric may not be an issue if the species detections are used to generate detection-corrected population estimates [18]. Our chosen sample size of 50 120-s clips is adequate for the objectives of this research. However, the limited sample size could be a potential reason that recall could not be calculated for multiple species. Therefore, studies that are focused on low vocalizing or rare species should consider using a larger sample size to create sufficient opportunities for detections.

As expected, we found that BirdNET performance improved when using a slightly higher confidence threshold of 0.25. At this level, $P_{C25}$ was $\geq$80% for 18 of the 20 focal species and the number of species with perfect precision scores ($P_{C25}$ = 1.000) increased from two species to eight species (Table 1). Overall, our measured BirdNET precision values were similar to the precision estimates of several studies reviewed by Pérez-Granados [19]. For example, Cole et al. [5] calculated an average BirdNET precision of 0.81 for 13 focal species in northwestern California. The high precision values calculated for focal species across most of these studies is likely due to selection bias for common regularly detected species. Common species with loud and/or distinct vocalizations typically have high estimates of precision (e.g., $P$ >0.8; [19]). The 20 focal species we selected for this study were common to grassland systems in Nebraska, and many of them have uniquely distinguishable calls, which are the likely reasons for our relatively high average performance metrics.

## Calculating species richness

The application of BirdNET to inform conservation and management decisions depends on the selection of the appropriate BirdNET confidence threshold. Confidence scores are not interchangeable between species and using the same confidence level may lead to different performance outcomes [18]. For this reason, previous efforts have focused on the development of species-specific confidence thresholds. For example, Cole et al. [5] used the maximum confidence score for false positive detections to determine the species-specific confidence threshold for 13 focal species. Bota et al. [11] used logistic regression to determine the confidence threshold for two target bird species as the minimum confidence score required to include detections with a 95% probability of correct identification. We similarly developed species-specific BirdNET confidence thresholds based on BirdNET performance in terms of precision (Table 2).

We were also interested in examining the influence of static BirdNET confidence thresholds on species richness measurements of the grassland bird community. Static confidence thresholds have been used in BirdNET applications to quantify broader measures of avian communities, such as species richness (e.g., [15]). We incorporated an SSC threshold with five static thresholds (0.1, 0.25, 0.5, 0.75, and 0.9) to examine the influence of these parameters on avian species richness estimates and compare these calculations to species richness estimates obtained through the point count surveys. The SSC thresholds ranged between 0.10 and 0.50, which were comparable to the lower end of the static thresholds. Based on the tradeoff between confidence and the number of BirdNET detections [10], it was not surprising for us to find that BirdNET-derived estimates of total species richness (focal and non-focal species) decreased as confidence level increased. In the short-duration, average BirdNET-derived estimates of species richness ranged between 2.95 and 7.130 species across all recorders at

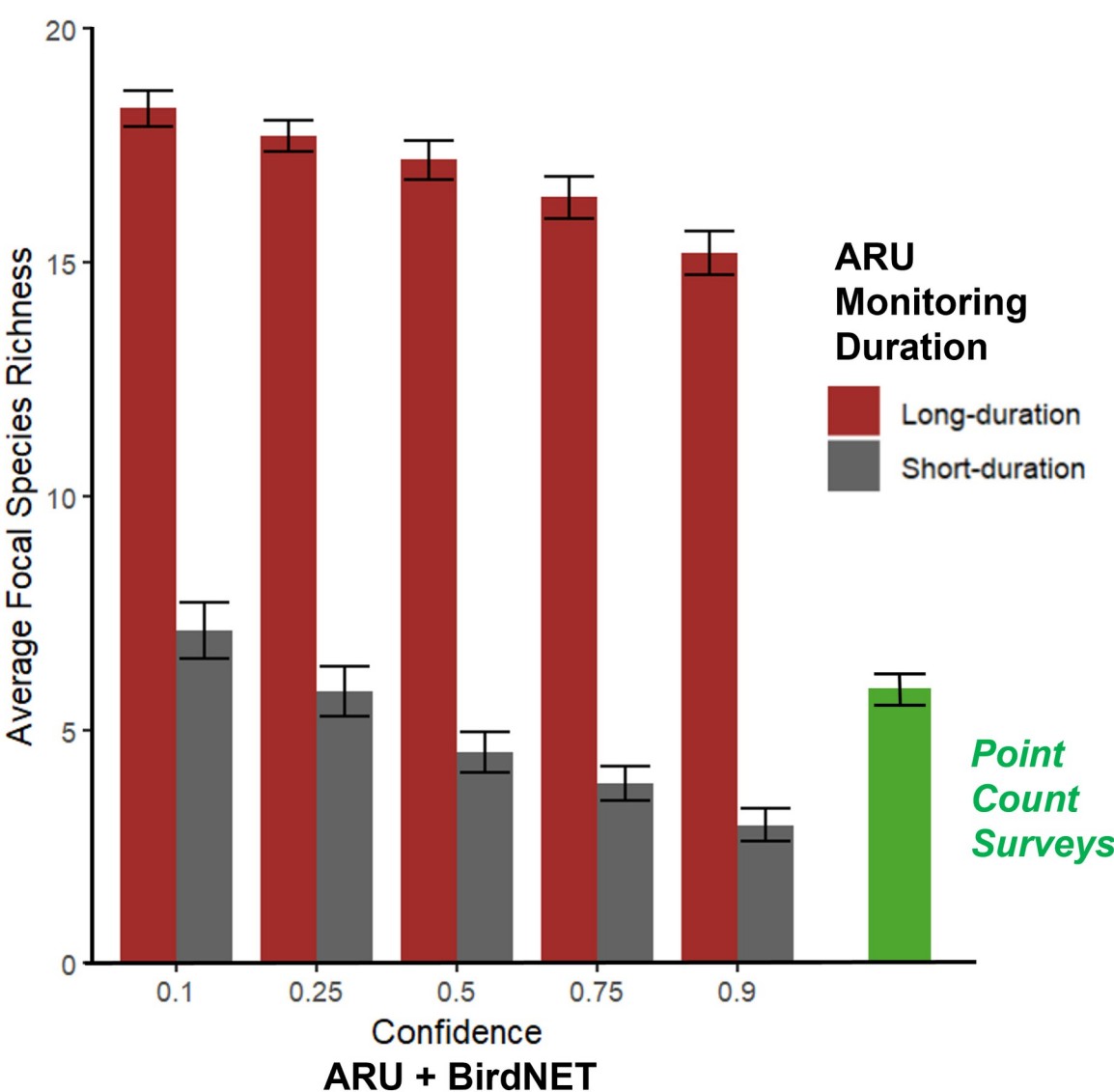

**Fig 4. BirdNET-derived estimates of average focal species richness across all recorders at static confidence levels of 0.1, 0.25, 0.5, 0.75, 0.9.** Short-duration measures of focal species richness were calculated based on the number of unique focal species detected by BirdNET from acoustic recordings within 1-hr of each point count survey. Long-duration measure of focal species richness was calculated based on the total number of focal species detected by BirdNET from all acoustic recordings from May 19th—July 19th, 2022.

confidence levels of 0.9 and 0.1, respectively. However, at the long-duration, average Bird-NET-derived estimates of species richness ranged between 15.2 and 18.3 at the same confidence levels (Fig 4).

## Comparing BirdNET to point count surveys

One of our primary objectives was to compare species richness estimates between point count surveys and ARUs. We observed or heard an average of 6.9 species at each point count location during the 60-day point count survey period. The comparability of species richness estimates varied by monitoring duration. In the short duration, BirdNET generated similar estimates of focal species richness as point count surveys when using lower confidence thresholds (SSC, 0.1, 0.25), but significantly lower focal species richness estimates using more conservative

confidence thresholds (≥0.50; Table 3). Interestingly, at the long duration period, BirdNET detected between six and nine more species per location, on average, across all confidence thresholds. Increasing our survey coverage (number of included recordings and surveys) allowed our ARU-based measures of species richness to increase to the point where 19 out of 20 focal species were detected within 15 days regardless of BirdNET confidence level (Fig 2B).

The only focal species that was never detected through BirdNET or point count surveys during the 60-day point count survey period was the Savannah Sparrow. This species was selected as a focal species because of its high number of detections by BirdNET in early spring (4,439 detections in March–April). However, we did not detect Savannah Sparrow using either method during the 60-day point count survey period beginning on May 19th, likely because this species migrated away from the study area by that time. This species is a common spring and fall migrant across the state of Nebraska, but it infrequently nests in the study area during the summer months [43–45]. We included this species as a focal species, in part, to exemplify the utility of BirdNET with long-term deployment of ARUs to monitor the phenology of migratory species.

Although several studies have highlighted the importance of performing SSC assessments [5, 11], our results indicate that an SSC threshold may be less important for long-duration community-level research objectives focused on metrics of species richness and diversity. At all confidence levels, BirdNET detected the same number of focal species after approximately two weeks of ARU deployment (Fig 2B), and each of these confidence thresholds produced greater long-duration estimates of focal species richness than manual point count surveys. Even at the most conservative BirdNET confidence threshold (0.90), we detected an average of six more focal bird species per site with the ARUs than the point count surveys (Table 3). For studies using long-term (>2 weeks) deployment of ARUs and where BirdNET performance has been validated for the bird community, our results suggest that selecting a moderate BirdNET confidence threshold of 0.50 or 0.75 would generate comparable estimates of species richness as the SSC threshold. Still, manual validation of BirdNET performance should be required for any ARU-BirdNET research application [18]. If BirdNET performance can be validated over a community of avian species (e.g., >80% overall precision), selecting a single confidence threshold could be a defensible and efficient way to generate community-level measures of species richness. While this approach may be effective for ARU applications at the community-level, studies focusing on a single bird species or rare species should focus on establishing SSC thresholds.

## Management implications

BirdNET has advanced tremendously in the past several years, enabling the detection of more species and at greater performance than earlier version releases. Despite these improvements, there are considerations that need to be made to optimize field research designs that leverage ARUs. The first consideration, which is not unique to BirdNET, is that PAM with ARUs inherently biases towards species that are more vocal than less vocal, cryptic species. The accuracy of BirdNET increases when analyzing bird vocalizations that BirdNET is more familiar with [20]. Second, PAM does not readily provide information on bird abundance or how birds utilize a particular site. As such, researchers should consider the behavior and call repertoire of each species prior to initiating a PAM study–either in a target species study or a community-level study. In addition, studies that combine PAM with traditional surveys such as point count surveys can help researchers collect information on relative abundance and behavior and produce more robust datasets to answer a variety of research questions [3, 30].

Our results also highlight the potential effectiveness of ARUs with BirdNET to monitor economically important game bird species such as Northern Bobwhite, Ring-necked Pheasant

(*Phasianus colchicus*), and Wild Turkey. BirdNET made between 4,083 and 34,608 detections of these species at default settings at our study site during the 5-month monitoring period spanning from March to July 2022. BirdNET performed noticeably well for these three species ($P_{C10} \geq 0.80$; F-score $>0.85$), likely due to their unique and distinguishable calls that are established in BirdNET's training datasets. We found BirdNET performance was similar to other automatic classifiers for these species [46, 47] and can outperform other neural network models designed to identify these species by their calls (e.g., [48]). Research opportunities exist to leverage PAM with automatic classifiers such as BirdNET to better understand occurrence patterns and vocalization phenologies of these and other grassland bird species.

## Supporting information

**S1 File. Details on point count surveys, summary of 50 randomly generated 120-s acoustic clips, summary table of all bird species detected by BirdNET, and photos of acoustic monitor installation in grassland and row-crop fields.**
(DOCX)

## Acknowledgments

We are grateful to Ryan Lamont, Hunter Swanson, and Wyatt Ervin for field assistance. We are also appreciative of the many private landowners that permitted site access for this study. Finally, we greatly appreciate the two anonymous reviewers and the associate editor that took time to provide feedback on earlier drafts.

## Author Contributions

**Conceptualization:** Andrew R. Little.

**Data curation:** Grace E. Schuster, Leroy J. Walston.

**Formal analysis:** Grace E. Schuster, Leroy J. Walston.

**Funding acquisition:** Andrew R. Little.

**Investigation:** Grace E. Schuster, Leroy J. Walston.

**Methodology:** Grace E. Schuster, Leroy J. Walston.

**Project administration:** Andrew R. Little.

**Software:** Grace E. Schuster, Leroy J. Walston.

**Validation:** Grace E. Schuster, Leroy J. Walston, Andrew R. Little.

**Visualization:** Grace E. Schuster.

**Writing – original draft:** Grace E. Schuster, Leroy J. Walston.

**Writing – review & editing:** Grace E. Schuster, Leroy J. Walston, Andrew R. Little.

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
