## [Decision Letter · Decision Letter 0]

30 Apr 2024

PONE-D-24-11491Evaluation of an Autonomous Acoustic Surveying Technique for Grassland Bird Communities in NebraskaPLOS ONE

Dear Dr. Schuster,

Thank you for submitting your manuscript to PLOS ONE. After careful consideration, we feel that it has merit but does not fully meet PLOS ONE’s publication criteria as it currently stands. Therefore, we invite you to submit a revised version of the manuscript that addresses the points raised during the review process.

Both reviewers have indicated minor revisions, and I agree with their opinion. Please make sure to address all their remaining concerns.

We look forward to receiving your revised manuscript.

Kind regards,

Luca Nelli, PhD

Academic Editor

PLOS ONE

Journal Requirements:

"Funding was provided by the US DOE Office of Energy Efficiency and Renewable Energy (EERE) Bioenergy Technologies Office under award DE-EE0009279. The views expressed in the article do not necessarily represent the views of the DOE or the US Government. The U.S. Government retains for itself, and others acting on its behalf, a paid-up nonexclusive, irrevocable worldwide license in said article to reproduce, prepare derivative works, distribute copies to the public, and perform publicly and display publicly, by or on behalf of the Government. We are grateful to Ryan Lamont, Hunter Swanson, and Wyatt Ervin for field assistance. We are also appreciative of the many private landowners that permitted site access for this study. Finally, we greatly appreciate the two anonymous reviewers and the associate editor that took time to provide feedback on earlier drafts."

"GS, LW, and AL received a Department of Energy grant to fund this work. The specific grant number is DE-EE0009279. No commercial companies funded this study or any of the authors. This is the URL to the funder website: https://www.energy.gov/eere/bioenergy/bioenergy-technologies-office

The funders did not play a role in study design, data collection and analysis, decision to publish, or preparation of the manuscript. "

3. We note that Figure 1 in your submission contain map/satellite images which may be copyrighted. All PLOS content is published under the Creative Commons Attribution License (CC BY 4.0), which means that the manuscript, images, and Supporting Information files will be freely available online, and any third party is permitted to access, download, copy, distribute, and use these materials in any way, even commercially, with proper attribution. For these reasons, we cannot publish previously copyrighted maps or satellite images created using proprietary data, such as Google software (Google Maps, Street View, and Earth). For more information, see our copyright guidelines: http://journals.plos.org/plosone/s/licenses-and-copyright.

Reviewers' comments:

Reviewer's Responses to Questions

**Comments to the Author**

1. Is the manuscript technically sound, and do the data support the conclusions?

Reviewer #1: Yes

Reviewer #2: Yes

2. Has the statistical analysis been performed appropriately and rigorously? 

Reviewer #1: Yes

Reviewer #2: Yes

3. Have the authors made all data underlying the findings in their manuscript fully available?

Reviewer #1: Yes

Reviewer #2: Yes

4. Is the manuscript presented in an intelligible fashion and written in standard English?

Reviewer #1: Yes

Reviewer #2: Yes

5. Review Comments to the Author

Reviewer #1: General comments

This is a very interesting, well written manuscript. It is very clear, aims are nicely defined, methods are appropriate, and results reflect aims. Nonetheless, I found some minor issues, listed below, to be addressed before publishing.

Specific comments

Introduction

1. Lines 105-106: as I guess, did you monitor breeding birds? If yes, please, add this detail because could be wintering birds.

Study area

2. Line 124: replace “acoustic monitors” with “ARUs”

3. Lines 128-129: did you placed ARUs in all these grassland types? Or only in one of them? Please specified.

Sampling design

4. Line 142: remove “separate”

5. Did you used some detection radius to count birds or did you count birds with “unlimited” distance (so, all birds seen/heard also at great distance)?

6. Line 156: replace “acoustic monitors” with “ARUs”.

7. Line 158: replace “space” with “distance”

8. Lines 162-163: why? To avoid the edge effect? If yes, please add this fact to be more clear.

Evaluating BIRDNET performance

9. Line 272: replace “to comparing” with “to compare”

Figures

10. Lines 391 and 393: replace Fig. A and Fig. B with Panel A and Panel B.

Reviewer #2: I have read the study with pleasure. It is a really good and valuable paper examining the effectiveness of a completely automatic acoustic approach to bird monitoring. The authors proved that by applying soundscape recording and automatic detection of bird vocalizations using BirdNet, they achieve the same or better level of effectiveness as human observers. They have really well-planned the study, estimated the recall and precision of automatic detections, and manually confirmed the detection of each species by BirdNet. In my opinion, the study should be published after minor revision.

Minor comments below:

L56-63: You should also point out some limitations. The first one is that PAM focuses only on vocalizing animals, thus we assume that vocalizing animals reflect overall biodiversity. The next issue is unequal detection range for differently vocalizing species (depends on frequency and amplitude of vocalisation). Additionally, biodiversity is not all; species abundance is also important but difficult to catch by PAM.

L99-104: The BirdNet does not detect signals on its own but compares the time period (3 sec) with the template. Thus, we may expect that the effectiveness of bird detection by BirdNet may differ between different bird communities and may vary depending on the time of day or season. Therefore, it is important to test the effectiveness of BirdNet under various ecological conditions. Your study does this, which is why it is important.

L123: Please provide more information about the timing of the breeding season within your study area. Additionally, could you share some basic information about species richness?

L159: Did you check the microphones before recording? Which gain setting did you apply?

L222: Is it not too low sample size obtain a good estimation of recall? Especially for rare or low-vocalizing species, you will get no detections.

6. PLOS authors have the option to publish the peer review history of their article (what does this mean?). If published, this will include your full peer review and any attached files.

Reviewer #1: No

Reviewer #2: No

---

## [Author Response · Author response to Decision Letter 0]

21 May 2024

Responses to Academic Editor Comments

Academic Editor Comment: “Please ensure that your manuscript meets PLOS ONE's style requirements, including those for file naming. The PLOS ONE style templates can be found at 

https://journals.plos.org/plosone/s/file?id=ba62/PLOSOne_formatting_sample_title_authors_affiliations.pdf”

Author’s Response: We formatted the revised manuscript to the best of our abilities to conform to the journal style requirements. 

Academic Editor Comment: “Thank you for stating the following in the Acknowledgments Section of your manuscript: "Funding was provided by the US DOE Office of Energy Efficiency and Renewable Energy (EERE) Bioenergy Technologies Office under award DE-EE0009279. The views expressed in the article do not necessarily represent the views of the DOE or the US Government. The U.S. Government retains for itself, and others acting on its behalf, a paid-up nonexclusive, irrevocable worldwide license in said article to reproduce, prepare derivative works, distribute copies to the public, and perform publicly and display publicly, by or on behalf of the Government. We are grateful to Ryan Lamont, Hunter Swanson, and Wyatt Ervin for field assistance. We are also appreciative of the many private landowners that permitted site access for this study. Finally, we greatly appreciate the two anonymous reviewers and the associate editor that took time to provide feedback on earlier drafts."”

"GS, LW, and AL received a Department of Energy grant to fund this work. The specific grant number is DE-EE0009279. No commercial companies funded this study or any of the authors. This is the URL to the funder website: https://www.energy.gov/eere/bioenergy/bioenergy-technologies-office

The funders did not play a role in study design, data collection and analysis, decision to publish, or preparation of the manuscript. "

Author’s Response: We removed the portion of the Acknowledgement section that described funding information and we included the updated funding statement within the cover letter. 

Academic Editor Comment: “We note that Figure 1 in your submission contain map/satellite images which may be copyrighted. All PLOS content is published under the Creative Commons Attribution License (CC BY 4.0), which means that the manuscript, images, and Supporting Information files will be freely available online, and any third party is permitted to access, download, copy, distribute, and use these materials in any way, even commercially, with proper attribution. For these reasons, we cannot publish previously copyrighted maps or satellite images created using proprietary data, such as Google software (Google Maps, Street View, and Earth). For more information, see our copyright guidelines: http://journals.plos.org/plosone/s/licenses-and-copyright.

Natural Earth (public domain): http://www.naturalearthdata.com/”

Author’s Response: Figure 1 was produced using ESRI World Imagery in ArcGIS Pro. Based on ESRI’s Terms of Use (e.g., https://doc.arcgis.com/en/arcgis-online/reference/static-maps.htm), we have permission to use static maps of ESRI World Imagery in academic publications such as this. We have updated the figure to credit ESRI for the source imagery. In addition, we confirmed that the use of the Basemap aligns with the conditions of the Creative Common Attribution 4.0 International license by reaching out to ESRI and obtaining a copyright permission letter which is attached to the revised submission.

Academic Editor Comment: “Please review your reference list to ensure that it is complete and correct. If you have cited papers that have been retracted, please include the rationale for doing so in the manuscript text, or remove these references and replace them with relevant current references. Any changes to the reference list should be mentioned in the rebuttal letter that accompanies your revised manuscript. If you need to cite a retracted article, indicate the article’s retracted status in the References list and also include a citation and full reference for the retraction notice.”

Author’s Response: We reviewed the reference list to ensure that all references are in the correct format. Three references were added to the reference list [13, 14, 27] and the order of the references was changed, but no references were removed. 

Responses to Reviewer #1 Comments

Reviewer 1 General Comment: “This is a very interesting, well written manuscript. It is very clear, aims are nicely defined, methods are appropriate, and results reflect aims. Nonetheless, I found some minor issues, listed below, to be addressed before publishing.”

Author's Response: We greatly appreciate the time you took to review our manuscript and provide feedback. 

Reviewer 1 Comment: “Lines 105-106: as I guess, did you monitor breeding birds? If yes, please, add this detail because could be wintering birds.”

Author’s Response: We focused on grassland birds during the breeding season, but the avian species included in this study consisted of both breeding grassland birds as well as year-round residents of Nebraska. This detail was added to the manuscript to enhance clarity. 

Reviewer 1 Comment: “Line 124: replace “acoustic monitors” with “ARUs””

Author’s Response: We abbreviated ‘acoustic monitors’ to ‘ARUs’.

Reviewer 1 Comment: “Lines 128-129: did you placed ARUs in all these grassland types? Or only in one of them? Please specified.”

Author’s Response: We did not specifically place ARU’s in these grassland types, they were just the dominant land types of this region. Instead we placed the monitors in Conservation Reserve Program (CRP) sites and to improve clarity, we did include the dominant grass species that were present at each of our grassland study sites.

Reviewer 1 Comment: “Line 142: remove “separate””

Author’s Response: The word ‘separate’ was removed. 

Reviewer 1 Comment: “Did you used some detection radius to count birds or did you count birds with “unlimited” distance (so, all birds seen/heard also at great distance)?”

Author’s Response: We are assuming that this comment is referring to our point count methods. During our point counts, we did not have a specific detection radius to count birds, mostly because the landscape is so open and flat you can hear birds from great distances with confidence. We did specify the lack of detection radius within our methods. 

Reviewer 1 Comment: “Line 156: replace “acoustic monitors” with “ARUs”.”

Author’s Response: We abbreviated ‘acoustic monitors’ to ‘ARUs’.

Reviewer 1 Comment: “Line 158: replace “space” with “distance””

Author’s Response: We replaced the word ‘space’ with ‘distance’.

Reviewer 1 Comment: “Lines 162-163: why? To avoid the edge effect? If yes, please add this fact to be more clear.”

Author’s Response: We made sure that the acoustic monitors were at least 100 meters away from the field edge not necessarily to avoid edge effect, but rather to make sure that the bird species detected was within our study site and not occupying a neighboring field. We added this detail to make the methods more clear. 

Reviewer 1 Comment: “Line 272: replace “to comparing” with “to compare””

Author’s Response: If this comment is referring to the sentence “In addition to comparing focal species richness, we compared total species richness estimates over the 60-day point count survey period” we feel that “to comparing” is the correct expression.

Reviewer 1 Comment: “Lines 391 and 393: replace Fig. A and Fig. B with Panel A and Panel B.”

Author’s Response: We replaced ‘Fig. A’ and ‘Fig. B’ with ‘Panel A’ and ‘Panel B’ in the revised manuscript.

Responses to Reviewer #2 Comments

Reviewer 2 General Comment: “I have read the study with pleasure. It is a really good and valuable paper examining the effectiveness of a completely automatic acoustic approach to bird monitoring. The authors proved that by applying soundscape recording and automatic detection of bird vocalizations using BirdNet, they achieve the same or better level of effectiveness as human observers. They have really well-planned the study, estimated the recall and precision of automatic detections, and manually confirmed the detection of each species by BirdNet. In my opinion, the study should be published after minor revision.”

Author’s Response: We greatly appreciate the time you took to review our manuscript and provide feedback. 

Reviewer 2 Comment: “L56-63: You should also point out some limitations. The first one is that PAM focuses only on vocalizing animals, thus we assume that vocalizing animals reflect overall biodiversity. The next issue is unequal detection range for differently vocalizing species (depends on frequency and amplitude of vocalisation). Additionally, biodiversity is not all; species abundance is also important but difficult to catch by PAM.”

Author’s Response: Limitations surrounding the challenges with only focusing on vocalizing animals, estimating species abundance using PAMs, and the potential difficulties with detecting some species that are less vocal were added. 

Reviewer 2 Comment: “L99-104: The BirdNet does not detect signals on its own but compares the time period (3 sec) with the template. Thus, we may expect that the effectiveness of bird detection by BirdNet may differ between different bird communities and may vary depending on the time of day or season. Therefore, it is important to test the effectiveness of BirdNet under various ecological conditions. Your study does this, which is why it is important.”

Author’s Response: Thank you for this comment. We agree.

Reviewer 2 Comment: “L123: Please provide more information about the timing of the breeding season within your study area. Additionally, could you share some basic information about species richness?”

Author’s Response: The timing of the grassland birds breeding season in our study area was added. Regarding species richness, we included a list of all confirmed species detections (ARUs and Point Counts) in the supplemental information. 

Reviewer 2 Comment: “L159: Did you check the microphones before recording? Which gain setting did you apply?”

Author’s Response: The gain setting was left at the default setting (18 dB) and microphones were checked before installation. We specified this in the new manuscript draft. 

Reviewer 2 Comment: “L222: Is it not too low sample size obtain a good estimation of recall? Especially for rare or low-vocalizing species, you will get no detections.”

Author’s Response: Fifty 120-second clips is an adequate sample size for estimating recall for our 20 common focal grassland bird species. Since we choose avian species that are highly vocal and make up the bulk of detections at each of our study sites, we felt it was likely we would detect each of these species within the timeframe. Unfortunately, due to choosing 120 random clips, we did not detect three focal species. While studies that are focusing on rare or low-vocalizing species should have a larger sample size, we feel the sample size we used allowed us to answer our research question. We made revisions in the discussion portion of the manuscript to clarify these shortcomings of calculating recall from low sample sizes.

---

## [Decision Letter · Decision Letter 1]

20 Jun 2024

Evaluation of an Autonomous Acoustic Surveying Technique for Grassland Bird Communities in Nebraska

PONE-D-24-11491R1

Dear Dr. Schuster,

We’re pleased to inform you that your manuscript has been judged scientifically suitable for publication and will be formally accepted for publication once it meets all outstanding technical requirements.

Kind regards,

Yuan Zhang, PhD

Academic Editor

PLOS ONE

Additional Editor Comments (optional):

Reviewers' comments:

Reviewer's Responses to Questions

**Comments to the Author**

1. If the authors have adequately addressed your comments raised in a previous round of review and you feel that this manuscript is now acceptable for publication, you may indicate that here to bypass the “Comments to the Author” section, enter your conflict of interest statement in the “Confidential to Editor” section, and submit your "Accept" recommendation.

Reviewer #1: All comments have been addressed

Reviewer #2: All comments have been addressed

2. Is the manuscript technically sound, and do the data support the conclusions?

Reviewer #1: Yes

Reviewer #2: Yes

3. Has the statistical analysis been performed appropriately and rigorously? 

Reviewer #1: Yes

Reviewer #2: Yes

4. Have the authors made all data underlying the findings in their manuscript fully available?

Reviewer #1: Yes

Reviewer #2: Yes

5. Is the manuscript presented in an intelligible fashion and written in standard English?

Reviewer #1: Yes

Reviewer #2: Yes

6. Review Comments to the Author

Reviewer #1: (No Response)

Reviewer #2: The authors considered all my comments and suggestions. I recommend publication of the manuscript. Congratulations on a really good paper!

7. PLOS authors have the option to publish the peer review history of their article (what does this mean?). If published, this will include your full peer review and any attached files.

Reviewer #1: No

Reviewer #2: No

---

## [Editor Report · Acceptance letter]

25 Jun 2024

PONE-D-24-11491R1 

PLOS ONE

Dear Dr. Schuster, 

I'm pleased to inform you that your manuscript has been deemed suitable for publication in PLOS ONE. Congratulations! Your manuscript is now being handed over to our production team.

Kind regards, 

on behalf of

Professor Yuan Zhang 

Academic Editor

PLOS ONE